# The Potential of *Nabis americoferus* and *Orius insidiosus* as Biological Control Agents of *Lygus lineolaris* in Strawberry Fields

**DOI:** 10.3390/insects14040385

**Published:** 2023-04-16

**Authors:** François Dumont, Mireia Solà, Caroline Provost, Eric Lucas

**Affiliations:** 1Centre de Recherche Agroalimentaire de Mirabel, 9850 Rue de Belle-Rivières, Mirabel, QC J7N 2X8, Canada; 2Laboratoire de Lutte biologique, Université du Québec à Montréal, Case Postale 8888, Succ. Centre-Ville, Montréal, QC H3C 3P8, Canada

**Keywords:** generalist predator, release rate, biological control, predation rate, omnivorous predator

## Abstract

**Simple Summary:**

The tarnished plant bug, *Lygus lineolaris*, is a major pest in eastern Canada strawberry fields. The objective of the present study was to evaluate the potential of two omnivorous predators of the pest: the damsel bug, *Nabis americoferus*, and the minute pirate bug, *Orius insidiosus*. The results in the laboratory showed that *N. americoferus* attacked all stages of the tarnished plant bug and, in the field, reduced its population for several weeks. Additionally, the impact was significant for every tested release period. Conversely, *O. insidiosus* only attacked smaller nymphs, and its effect was marginal. These results lead the way for an effective biological control strategy based on the use of *N. americoferus* against the tarnished plant bug.

**Abstract:**

The tarnished plant bug, *Lygus lineolaris*, is a major strawberry pest. Only marginally effective control methods exist to manage this pest. Various predators attack *L. lineolaris*, but their potential is overlooked. In this study, we explore the potential of two omnivorous predators of the tarnished plant bug: the damsel bug, *Nabis americoferus*, and the minute pirate bug, *Orius insidiosus*. Firstly, the predation rate of these predators was measured in laboratory tests. Secondly, their potential release rates and release periods were determined in the field using strawberry plants. The results show that *N. americoferus* feeds on all nymphal stages and adults of the tarnished plant bug, while *O. insidiosus* attacks only smaller nymphs (up to the N2 stage). In the field, all tested densities of *N. americoferus* (0.25, 0.5, and 0.75 individual/plant) reduced the population of the tarnished plant bug for several weeks compared with the control treatment, but the effect of *O. insidiosus* alone was marginal. Additionally, for all the release periods tested, *Nabis americoferus* was efficient in reducing the pest population. These results demonstrate the potential of *N. americoferus* to control the tarnished plant bug in strawberry fields. We discuss the possible application of these results for establishing an effective and economically viable biological control strategy.

## 1. Introduction

The tarnished plant bug (TPB), *Lygus lineolaris* (Palisot de Beauvois) (Hemiptera: Miridae), is a major pest in strawberry fields [1]. This omnivorous insect feeds on more than 350 plant species, including over 120 economically important crops (e.g., cotton, alfalfa, soybeans, apples, strawberries, and buckwheat) [2,3,4]. Pest control strategies against TPB mainly rely on the use of broad-spectrum insecticides [5], which threaten non-targeted beneficial organisms [6,7]. Alternatively, several biological control agents (e.g., entomopathogenic fungi, parasitoids, and predators) may contribute to the control of TPB populations [8,9,10,11,12,13]. However, the role of the natural enemies of the TPB has been mainly overlooked. The potential of these organisms is yet unexploited and could constitute a part of a non-chemical solution for the management of TPB in organic strawberry fields.

The guild of TPB predators is mainly composed of predatory hemipterans and spiders [11,14,15,16]. For instance, Arnoldi et al. [16] observed that the hemipterans *Nabicula subcoleoptrata* (Kirby) (Hemiptera: Nabidae), *Zelus socius* (Stal) (Hemiptera: Reduviidae), *Phymata pennsylvanica* (Handlirsch) (Hemiptera: Reduviidae), and *Podisus maculiventris* (Say) (Hemiptera: Pentatomidae) consumes up to three TPB adults per day. Additionally, predatory hemipterans belonging to the Nabidae, Geocoridae, and Anthocoridae have been revealed as potentially important predators of TPB [8,11,17]. Hagler et al. [11] observed that about 20% of *Nabis alternatus* (Parshley) (Hemiptera: Nabidae), *Geocoris punctipes* (Say) (Hemiptera: Geocoridae), and *Orius tristicolor* (White) (Hemiptera: Anthocoridae) captured in organic strawberry fields had traces of the western tarnished plant bug, *Lygus hesperus* (Knight) (Hemiptera; Miridae) in their guts. Among them, *O. tristicolor* was the most abundant predator, representing 63.7% of all the TPB predators captured in the study. In eastern Canada, the common damsel bug, *Nabis americoferus* (Carayon) (Hemiptera: Nabidae), and the minute pirate bug, *Orius insidiosus* (Say) (Hemiptera: Anthocoridae), have been regularly found preying on TPB (F. Dumont, personal observation). Moreover, Pumariño et al. (2011) reported that females of both predators laid more eggs in the presence of their competitor than in the presence of conspecifics [18]. Therefore, releasing both predators in a biological control program may increase the numerical reproductive response and consequently the overall impact on the TPB. The same authors suggested that the simultaneous use of an ambush predator, such as *N. americoferus*, and an active-searching predator, such as *O. insidiosus*, may generate a facilitation effect [18]. Yet, the action of the active-searching predator would increase the focal prey movement and then increase the encounter opportunity with the ambush predator [19].

The potential of damsel bugs as predators of TPB has been already investigated [17]. It has been shown that even the smallest nymphal stages of *N. alternatus* can feed on small TPB nymphs [17]. Additionally, similar to TPB [20,21], damsel bugs hibernate as adults close to autumnal hosts (Perkins 1971; F. Dumont, personal observations). *Nabis* species are omnivorous predators and feed on several prey species, including aphids, leafminers, and plant bugs. Hence, they may play a role in the biological control of several pest species and maintain their population in agroecosystems in the absence of the targeted species. *Nabis americoferus* are highly fertile; females lay an average of 160 eggs and up to 250 eggs in about three weeks under laboratory optimal conditions [22]. These characteristics suggest that *N. americoferus* could be a major contributor to the management of the TPB in strawberry fields.

The commercially available minute pirate bug, *O. insidiosus*, is a small but voracious predator. This omnivorous species feed on several prey species, including thrips, plant bugs, aphids, and mites, as well as plant resources (mainly pollen) [23]. Its life cycle is short (about 21 days from egg to egg), and its fertility is modest (an average of 30 eggs per female) [24,25,26]. By feeding on the smaller and more cryptic TPB nymphal stages, *O. insidiosus* could constitute an important complement to *N. americoferus* to manage TPB populations in strawberry fields.

This study aims to determine the potential of *N. americoferus* as a biological control agent of TPB and evaluate the contribution of *O. insidiosus*, as a potential complementary agent to *N. americoferus*, in strawberry fields. First, the predation rate of both predators for various developmental stages of TPB was evaluated in the laboratory. Second, *N. americoferus* and *O. insidiosus* were released in the field to determine the optimal release rate, the release period, and the contribution of *O. insidiosus*.

## 2. Materials and Methods

### 2.1. Insect Rearing

The *Nabis americoferus* species used for laboratory and field trials were originally collected from the field at various locations of Quebec (mainly in the Laurentian region) and were then reared in the laboratory on eggplants, *Solanum melongena* L. (Solanaceae), and fed ad libitum with green peach aphid, *Myzus persicae* (Sulzer) (Hemiptera: Aphididae).

The *Orius insidiosus* species used for field trials were supplied by Anatis Bioprotection Inc. (Quebec). Alternatively, the *O. insidiosus* species tested in the laboratory came from a local rearing facility established by the same commercial source. The population at the local rearing facility was provided with romaine lettuce as oviposition sites and was fed with *Ephestia kuehniella* (Zeller) (Lepidoptera: Pyralidae) eggs. *Lygus lineolaris*, originally collected from fields at various locations of Quebec (mainly in the Laurentian region), were reared on romaine lettuce leaves that provide both food and oviposition substrate. New leaves were provided every four days. All insect-rearing procedures were performed at 25 °C, about 60% RH, and 16L:8D.

### 2.2. Laboratory Experiments

*Nabis americoferus* adults and N3 and N5 nymphs were matched with the development stages of N3, N5, and adult TPB. For this purpose, insects were introduced in a plastic box (14 × 14 × 7 cm) with a single strawberry leaf at the center. The petiole of the leaf was embedded in a little container with water. The lid and sides of the box were holed and closed with a fine mesh to allow air ventilation. Before the test, the predator was starved for 24 h, having only access to a leaf of lettuce as a source of water. In each test, a single predator was allowed to feed on 15 prey TPB offered at the same developmental stage. After 24 h, the predator was removed, and the prey still alive were counted, assuming the others had died. Control treatments consisted of TPB at each development stage tested and held for 24 h without predators.

The same experiment was used to test the level of predation of *O. insidiosus* adults and nymphs (N3 and N5) on TPB nymphs (N2 and N3). For each treatment combination and predator species, 15 replicates were performed.

The conditions in all insect-rearing procedures were maintained at 25 °C, about 60% RH, and 16L:8D.

### 2.3. Field Experiments

Field tests were carried out at the research experimental farm in Mirabel (Mirabel, QC, Canada) (Lat. 45,659,450; long. −74,080,897). In early June, 96 plots of 2.5 × 1 m were defined. A distance of ten meters (lengthwise and crosswise) was kept among the plots. Each plot included 16 strawberry plants (Albion variety). Wild plants, red-rooted pigweed (*Amaranthus retroflexus* (Amaranthaceae)) and ragweed (*Ambrosia trifida* (Asteraceae)) were preserved near the black plastic mounds (about 1 m lengthwise), while the other plants were regularly cut. We left these wild plants to ensure a sufficient TPB population to observe the effect of our treatments. All three experiments (*Nabis* release rate, *Nabis* release periods, and *Orius* contribution) were conducted in the same field. The treatment applied to the plots was randomly assigned.

#### 2.3.1. *Nabis americoferus* Release Rate

Three release rates (treatments) of *N. americoferus* were compared: 0.25, 0.50, and 0.75 individuals per strawberry plant. A control treatment without predator release was also implemented. For all treatments, *N. americoferus* at a nymphal instar N5 and/or young sexually immature adults (less than 5 days old) from laboratory rearing were released when TPB nymphs were first found in the field (8 August 2019).

#### 2.3.2. *Nabis americoferus* Release Periods

Three release periods were tested: (1) at the 1st observation of TPB adults on strawberry plants (1st generation adults) (26 July 2019); (2) at the 1st observation of TPB nymphs (8 August 2019); and (3) at the 1st observation of 2nd generation adults (26 August 2019). In each case, a rate of 0.5 *N. americoferus* at the N5 stage and/or sexually immature adults per strawberry plant was used.

#### 2.3.3. *Orius insidiosus* Contribution

Four treatments were compared: (1) control treatment without predator; (2) *O. insidiosus* used alone; (3) *O. insidiosus* and *N. americoferus* used in combination, and (4) *N. americoferus* used alone. When *O. insidiosus* was introduced alone, two releases were performed following the distributor’s recommendations. These were on 1 and 9 August 2019 at the rate of 1.56 adults per strawberry plant. *Orius insidiosus* was released on these dates because most TPBs were at young nymphal stages (N1 and N2). For the treatment where both predators were introduced in combination, *O. insidiosus* was introduced on 1 August and *N. americoferus* on 8 August. The same release date was used when *N. americoferus* was introduced alone. In both treatments, *N. americoferus* was introduced at a rate of 0.25 individuals per plant.

Twelve replicates (plots) of each treatment were performed for each experiment.

#### 2.3.4. TPB Monitoring

The TPB population was assessed by counting the nymphs and adults obtained from the weekly beating sheet sampling of two random strawberry plants per plot over the season (24 July to 17 September). Overall, 1710 strawberry plants were sampled. The beating sheet-sampling process involved hitting the plant three times with the hand while a white cloth was placed below the sampled plant.

### 2.4. Statistical Analysis

The predation rates in the laboratory experiment, either of *N. americoferus* or *O. insidiosus* on the TPB, were assessed using generalized linear models (GLMs) for Poisson distributed data. Independent models for each TPB developmental stage were performed. Each model included the predator’s development stage as an explanatory variable.

The effect of *N. americoferus* release rate and release period as well as the effect of *O. insidiosus* on the TPB population on strawberry plants were assessed using a generalized linear mixed model (GLMER) for negative binomial distributed data. The first model included the treatment (*N. americoferus* release rate), the date of observations (linear and quadratic), and their interaction as fixed effects. The models on the release rate included observations carried out after the release of predators.

Three models were run to test the *N. americoferus* release period on the TPB population (one for each release period). These models only included observations for a period of three weeks after the release of *N. americoferus*. The treatment (control vs. *N. americoferus*) and the date were included as fixed effects.

The model to test the effect of *O. insidiosus* on the TPB population on strawberry plants included the treatment (*O. insidiosus* alone, *N. americoferus* alone, and a mixture of both species), and the date as fixed effects. This model, as well as the model to test the effect of *N. americoferus* release rate, included observations made between 13 August and 17 September, inclusively.

We used the same control treatments in all models.

For all the models, the plot was included as a random effect and the statistical significance (α = 0.05) of explanatory variables was estimated with the likelihood ratio test (LRT) using the function *drop1* in R. Differences among treatments were assessed with Tukey’s test using the *glht* function (package *multcomp* [27]). All analyses were performed using R [28].

## 3. Results

### 3.1. Predation Rate

#### 3.1.1. Predation of *Nabis americoferus* on TPB

Among all the tested *N. americoferus* developmental stages, only *N. americoferus* adults had a significant effect on the number of dead TPB adults (LRT_3_ = 24.09; *p* < 0.0001) (Figure 1A). On the other hand, *N. americoferus* at the adult and N5 nymphal stages significantly increased the number of dead N5 TPB, whereas *N. americoferus* at the N3 stage presented no differences in the number of dead TPB from the control treatment (LRT_3_ = 18.52; *p* = 0.0003) (Figure 1B).

All the tested stages of *N. americoferus* significantly killed the TPB at the N3 stage (LRT_3_ = 103.11; *p* < 0.0001) (Figure 1C). Moreover, *N. americoferus* adults killed significantly more TPB at the N3 stage than *N. americoferus* at the N3 and N5 nymphal stages (Figure 1C).

#### 3.1.2. Predation of *Orius insidiosus* on TPB

All the tested development stages of *O. insidiosus* (N3, N5, and adults) significantly increased the number of dead TPB nymphs at the N2 stage (LRT_3_ = 43.20; *p* < 0.0001) (Figure 2A), whereas TPB at the N3 nymphal stage was not significantly killed by *O. insidiosus* (LRT_3_ = 2.59; *p* = 0.46) (Figure 2B).

### 3.2. Field Tests

#### 3.2.1. *Nabis americoferus* Release Rates

The mean number of TPB individuals (all stages pooled) per strawberry plant in the control treatment was 0.7 (± 0.1) over the whole sampling period (from the middle of July to the middle of September). All the released rates of *N. americoferus* had a significant negative impact on the TPB population (LRT_3_ = 13.30; *p* = 0.004) (Figure 3). On average, the TPB population in the control treatment was 2.3 (95% CI: [1.4–4.1]), 4.2 ([2.4–7.8]), and 2.6 ([1.6–4.5]) times higher than in the plots with low, medium, and high rates of *N. americoferus*, respectively. The three *N. americoferus* treatments were not statistically different from each other over the whole sampling period. The TPB population increased over the season following a quadratic relationship (LRT_1_ = 24.01; *p* < 0.0001). The interaction between the release rate and the observation date was not significant (LRT_6_ = 3.77; *p* = 0.71).

#### 3.2.2. *Nabis americoferus* Release Periods

From 30 July to 13 August, the mean number of TPB in the control treatment was 0.2 (±0.05) individuals per strawberry plant. The first *N. americoferus* release period (corresponding to the observation of the first-generation TPB adults) had a significant negative impact on the TPB population (LRT_1_ = 4.61, *p* = 0.03) (Figure 4).

From 13 August to 27 August, the mean number of TPB per strawberry plant was 0.3 (±0.08) in the control treatment, which is 3.5 (95% CI: [1.33–16.5]) times higher than in the plots with *N. americoferus* released during the second release period (corresponding to the observation of the first TPB nymphs). Hence, *N. americoferus* released during the second release period significantly reduced the TPB population (LRT_1_ = 4.11; *p* = 0.04) (Figure 4).

From 27 August to 17 September, the mean number of TPB per strawberry plant in the control treatment was 0.9 (±0.14). The *N. americoferus* at the third release period (observation of second-generation TPB adults) significantly reduced the TPB population (LRT_1_ = 3.92; *p* = 0.04) (Figure 4). On average, the TPB population in the control treatment was 2.2 (95% CI: [1.4; 4.4]) times higher than in the treated plots. Moreover, in this third release period of *N. americoferus*, the TPB population did not significantly increase in either treatment (LRT_1_ = 2.41; *p* = 0.12) (Figure 4).

#### 3.2.3. Contribution of *Orius insidiosus*

Both treatments with *N. americoferus* (with or without *O. insidiosus*) negatively impacted TPB’s population (LRT_3_ = 7.95; *p* = 0.04) (Figure 5). However, the release of *O. insidiosus* alone did not significantly reduce TPB’s population. The mean number of TPB per plant was 1.6 (95% CI: [1.0; 2.7]), 3.0 ([1.6; 6.7]), and 2.3 (95% CI: [1.4; 4.1]) higher in the control treatment than in the treatments with singly *O. insidiosus*, *O. insidiosus*, and *N. americoferus* and singly with *N. americoferus*, respectively.

## 4. Discussion

The predation rates of *N. americoferus* on all the tested stages of TPB demonstrate its potential as a biological control agent against this pest. Each stage of *N. americoferus* can kill the equivalent or younger stages of TPB. This indicates that most *N. americoferus* stages can contribute to the control of TPB. Additionally, our laboratory results show that *O. insidiosus* has the potential to contribute to the control of TPB at the N2 nymphal stage and presumably (although not tested) also to younger ones (N1). This potential of *N. americoferus* was confirmed in field experiments on strawberries, but the effect of *O. insidiosus* on the TPB population was marginal and not significant. Then, in a biological control strategy against TPB, *N. americoferus* may play a main role in rapidly (from two weeks after release) and sustainably reducing the populations of the pest, while *O. insidiosus* may play a role as a secondary (and complementary) biological control agent since its contribution is limited.

The size ratio between predators and their prey is decisive in the outcome of a predator–prey encounter [29]. Our laboratory results showed that smaller and younger stages of TPB (N3 and presumably N1 and N2) are very vulnerable to predation by both *N. americoferus* and *O. insidiosus*. Therefore, synchronization between the developmental stages of *N. americoferus* and TPB will be decisive for the effectiveness of biological control by the predator, given that in this study, *N. americoferus* was shown to be able to affect prey that were at most at about the same stage of development. Under Canadian conditions, overwintering TPB species become active in April and/or the first weeks of May [30], while overwintering *N. americoferus* species are found active about a week or two after TPB (F. Dumont, personal observation). Hence, the natural life cycle of *N. americoferus* seems to be slightly behind that of TPB. Moreover, at a temperature of 27 °C, the nymphal development of *N. americoferus* takes an average of 24 days to be completed [31], compared with 18.8 days for TPB [32]. This suggests that the synchronicity between *N. americoferus* and TPB in spring may favor the prey over the predator. Consequently, the period of the year when *N. americoferus* effectively and naturally regulates the populations of TPB could be limited to some weeks during summer and fall (most *N. americoferus* are adults at that time). As a result, *N. americoferus* would not naturally protect strawberry plants during the whole production season (mainly June for summer-bearing strawberry plants and from midsummer to early autumn for everbearing plants). The natural regulation of TPB is certainly not to be overlooked, especially since the TPB and *N. americoferus* use the same autumnal hosts (e.g., mullein plants) [33]. However, the natural contribution of *N. americoferus* to TPB’s biological control during both summer and fall remains to be determined. Finally, the effectiveness of biological control by *N. americoferus* can vary from year to year depending on weather conditions and demographic factors of the prey population. Since our study was only conducted over one summer, future studies will increase our knowledge of this system.

Thus, well-synchronized releases of *N. americoferus* to the TPB cycle could constitute an effective augmentative biocontrol strategy. From an augmentative biocontrol perspective, such as our field experiment, the predators were released depending on the developmental stage of the TPB. *Nabis americoferus* adults were efficient in reducing the TPB population regardless of the developmental stage of the TPB. Therefore, the window for controlling the TPB is wide when *N. americoferus* is released at the adult stage, and the releases of *N. americoferus* could be synchronized with the period of vulnerability of strawberries. In our experiment, *N. americoferus* was shown as a promising biological control agent of the TPB when it was released at either 0.25, 0.5, or 0.75 individuals per strawberry plant. We released mainly adults and measured their local effect on the TPB population. However, these *N. americoferus* may have dispersed shortly after their release. The release of *N. americoferus* nymphs could extend the local effect, as these nymphs have a more limited dispersal capacity. Thus, when young nymphal stages (N1 and N2) of the TPB are first observed in the field, the release of N3–N4 nymphs of *N. americoferus* could have an optimal impact on the pest’s population.

Our results indicate that *Orius insidiosus* feeds on the TPB, but it is not an effective biological control agent against this pest. Hagler et al. [11] observed that up to 16.4% of captured *O. tristicolor* in strawberry fields had TPB in their gut, a rate comparable to what they determined for *G. punctipes* (18.2%) and lower than what they determined for *N. alternatus* (33.3%). That level seems not to be enough for the effective control of the TPB population. Moreover, our results show that *N. americoferus* and *O. insidiosus* did not have a synergetic effect on TPB’s population. The expected facilitation effect between the active searching *O. insidiosus* and ambush predator *N. americoferus* was not observed. However, *O. insidiosus* was not harmful to *N. americoferus*, which maintained its effectiveness even in the presence of *Orius*. Thus, a management strategy favorable to *O. insidiosus* and other predators, from the perspective of biological control through conservation, could be an interesting strategy. Theoretically, the TPB could respond to predation risk and modulate feeding and egg-laying behavior [34]. The anti-predatory behaviors (e.g., escape, camouflage, use of refuge, etc.) of the TPB, however, remain little known.

Massive releases of *N. americoferus* over the entire area of a strawberry field is a strategy that seems economically unviable. Thus, biological control based on *N. americoferus* would require a combination of strategies that reduce the extent of the surface to be treated. First, TPB has been shown to aggregate in trap crops, where they can be targeted by repressive treatments [33,35,36,37]. For instance, Swezey et al. [37] reported that approximately 90% of the TPB population was concentrated in trap crops and adjacent strawberry rows (covering only 6% of the area). Hence, a large part of the TPB population could be targeted while minimizing the number of *N. americoferus* to be released. Moreover, Hagler et al. [11] observed that the rate of predation by *N. alternatus* was higher in the alfalfa trap crops than in the strawberry plants. This increased performance of *N. alternatus* in alfalfa could be explained by its type of predation. *Nabis* spp. are ambush predators waiting for their prey. In alfalfa plots, where *L. hesperus* populations were denser, the rate of the encounter between *N. alternatus* and its prey was certainly higher [11]. As a result, the rate of predation by *N. alternatus* was increased. Hence, the combination of trap crop and predator release may have a synergetic effect.

## Figures and Tables

**Figure 1 insects-14-00385-f001:**
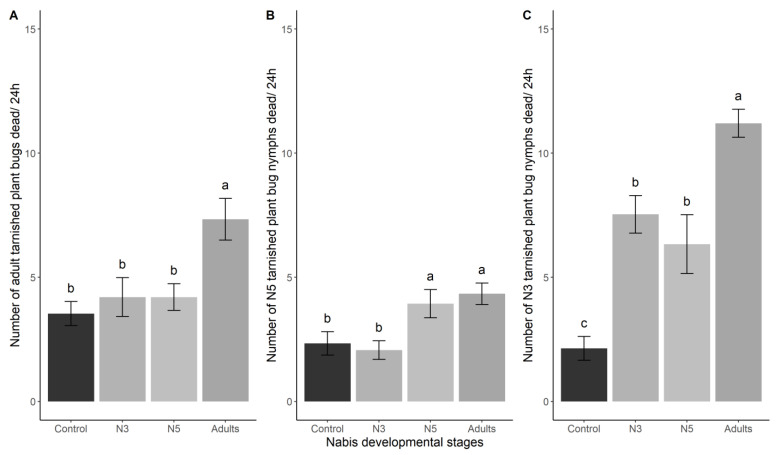
Mean mortality of TPB adults (**A**), N5 (**B**), or N3 (**C**) nymphs as a function of the *Nabis americoferus* developmental stage. Error bars represent standard errors. Letters correspond to significant differences among treatments (α = 0.05).

**Figure 2 insects-14-00385-f002:**
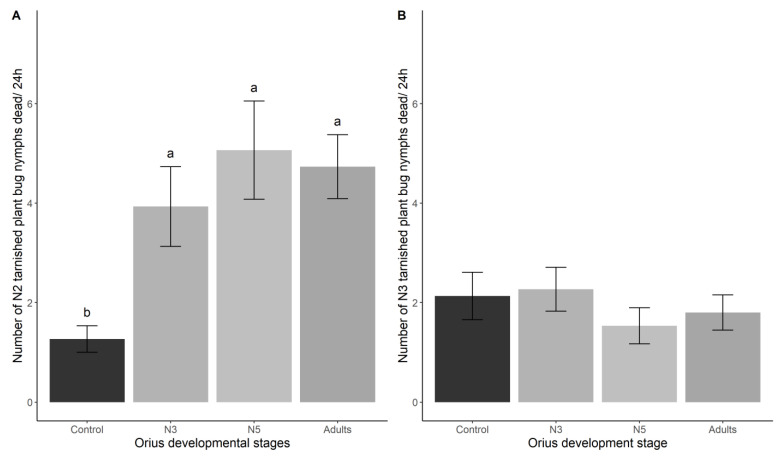
Mean mortality of TPB N2 (**A**) and N3 (**B**) nymphal stages as a function of the *Orius insidiosus* developmental stage. Error bars represent standard errors. Letters correspond to significant differences among treatments (α = 0.05).

**Figure 3 insects-14-00385-f003:**
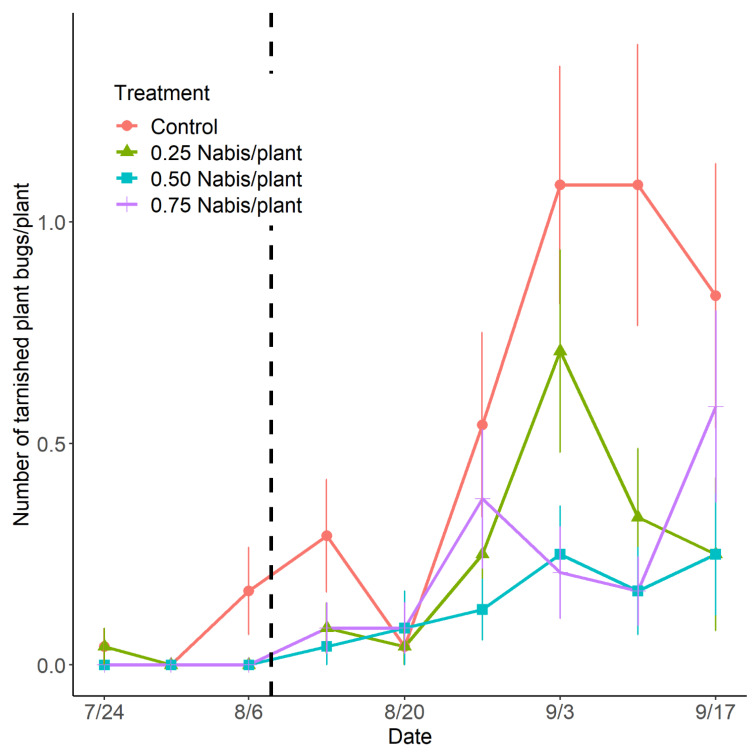
Seasonal variation in the tarnished plant bug population based on the rate of release of *Nabis americoferus*. The vertical line refers to the date of the predator’s release (8 August). The error bars correspond to standard errors.

**Figure 4 insects-14-00385-f004:**
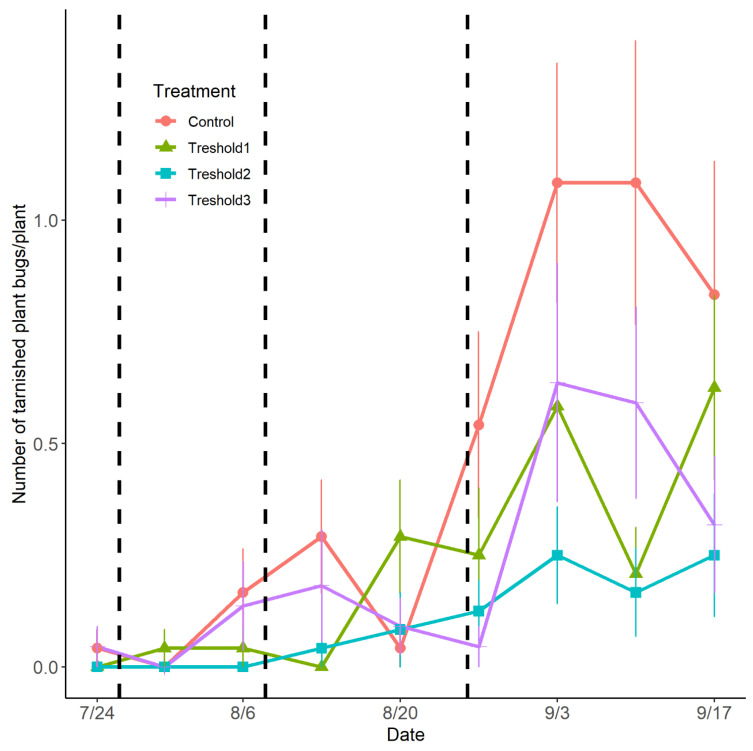
Seasonal variation in the population of the tarnished plant bug observed on strawberry plants depending on the release periods (release rate of 0.5 *Nabis americoferus* per strawberry plant). The vertical lines refer to the dates of the release (25 July, 7 August, and 25 August). The error bars correspond to standard errors.

**Figure 5 insects-14-00385-f005:**
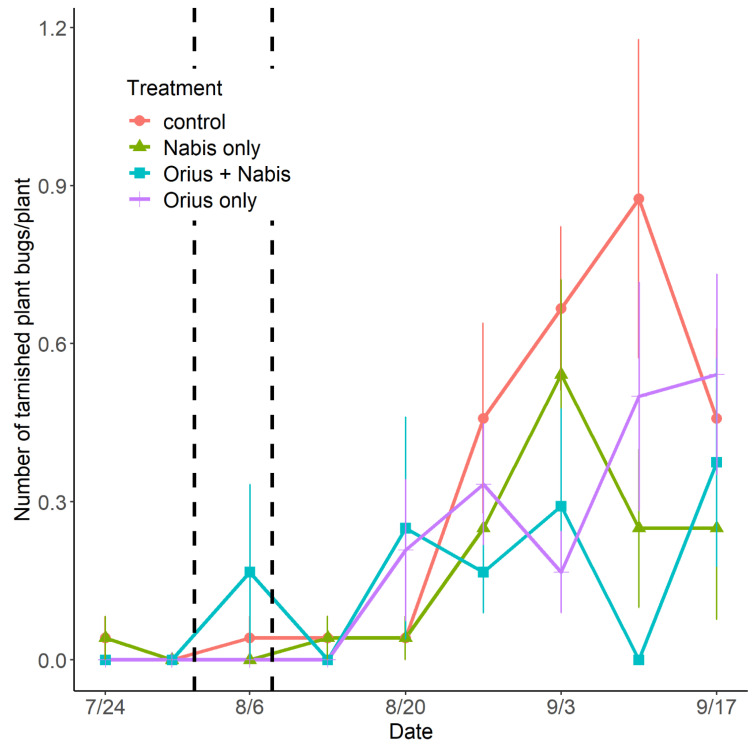
Seasonal variation in the population of the tarnished plant bug observed on strawberry plants depending on the *Orius insidiosus* and *Nabis americoferus* treatments. The vertical lines refer to the dates of the release of *Orius insidiosus* (1 August and 7 August). The error bars correspond to standard errors.

## Data Availability

The data are openly available on Zenodo (10.5281/zenodo.7411574).

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
