# Peer review of "The Potential of *Nabis americoferus* and *Orius insidiosus* as Biological Control Agents of *Lygus lineolaris* in Strawberry Fields"

_insects, 2023, doi:10.3390/insects14040385_

Round 1

Reviewer 1 Report

This manuscript would really benefit from replicating field experiments a second year, because it's really difficult to make conclusions.

This manuscript reports on laboratory and field experiments that investigated the potential of a damselfly (Nabis americoferus) and the insidious flower bug (Orius insidiosus) as biological control agents for the tarnished plant bug (TPB) in strawberries. Laboratory experiments demonstrated that Nabis, and to a lesser extent O. insidious, do predate on TPB. Field experiments, which included different rates and release timing of Nabis, as well as Orius alone and in combination with Nabis, were less convincing.

Although statistically significant differences are reported, it’s not clear what specifically is significantly different. What data were compared – season total average counts, and if so is this an appropriate comparison? The large variances and overlapping SEMs makes it difficult to really see where differences occurred – perhaps later in the season when TPB counts were highest (9/3 and 9/10). With 12 replicates it is quite easy to detect statistically significant differences, but what this means biologically is difficult to assess. This is compounded by the fact that TPB populations were very low and variance was high.

There was a time when most journals required that field studies be replicated over two years, or at least be replicated over several locations in one year. For some reason many have drifted away from this standard, which is unfortunate, because results can vary with pest pressure and weather conditions.

Specific comments that would add to the manuscript are listed below.

Editorial Comments: There many grammatical errors throughout the manuscript. A review by someone with English as a first language before submitting would have been helpful.

Introduction: The introduction could be shortened with more concise writing. It is not necessary to list all of the predators previously reported, nor is it necessary to provide such detailed information on predator development time or fecundity.

Line 28: Is there a reference to document TPB is omnivorous? I have not heard this before.

Line 50: The official common name of Orius insidiosus is the insidious flower bug, not minute pirate bug.

Line 73: Please provide a reference that Orius feeds on pollen. Why (and how) would an insect with sucking mouthparts feed on pollen

Insect Rearing: how long were these colonies maintained in the laboratory?

Line 101-102: Not really sure what is being described here – what insect’s access was prevented and from what?

Line 118: What other plants were regularly, and how were they cut.

Line 146: The design of the experiment needs to be explained to indicated that certain treatments were double-dipped – i.e., the same control plots (and probably 2 others) were used in the analysis of all 3 experiments. It’s confusing, because a total of 10 treatments across all 3 experiments are described, and with 12 replicates this equals 120 plots, but there were only 96 plots (line 114)

Line 203: This pertains to the beginning of the each experiment results sections. The mean number of TPB per plant in the control is listed as 0.65. It’s not clear what this value represents – perhaps the average density across all sample dates?  If so, it is a meaningless value that has no relevance. What is more important is a description of TPB phenology across the study period.

Line 309: what is anti-predatory behavior?

Fig. 3-5: The color selection is not good for colorblind readers to distinguish treatments. In fact there’s really no need for different colors, as different shaped or filled and not filled data points along with dashed and solid lines could be used. Also are the vertical bars on data points standard error of mean?

The “Conclusion” is actually a Discussion and should be referred to as that. Also, there should be some discussion of how the results relate to application, and what next avenues of research are necessary. The last couple of sentences end rather abruptly with no conclusion.

Author Response

  1. A) This manuscript would really benefit from replicating field experiments a second year, because it's really difficult to make conclusions.

*We do not have a second year of observation, but this does not invalidate our conclusions and we have performed enough replications to sustain the results.

  1. B) Although statistically significant differences are reported, it’s not clear what specifically is significantly different. What data were compared – season total average counts, and if so is this an appropriate comparison? The large variances and overlapping SEMs makes it difficult to really see where differences occurred – perhaps later in the season when TPB counts were highest (9/3 and 9/10). With 12 replicates it is quite easy to detect statistically significant differences, but what this means biologically is difficult to assess. This is compounded by the fact that TPB populations were very low and variance was high.

*Details of our statistical models are provided in the methodology section.  We compared average populations of TPB after the release of predators according to treatments.  Models control for temporal variations. Figures describe seasonal variations well and differences between treatments are also observable.  In our experiment, we measured the local effect of introducing predators. This is how we interpret the results in the discussion. Finally, the TPB’s population densities were not low as the reviewer suggests. They are normal.

  1. C) There was a time when most journals required that field studies be replicated over two years, or at least be replicated over several locations in one year. For some reason many have drifted away from this standard, which is unfortunate, because results can vary with pest pressure and weather conditions.

*It is true that variations in pest populations or weather conditions can influence the results.  However, our results and conclusions remain valid and are statistically supported. The results are a contribution to understanding the potential of Nabis and Orius as biological control against TPB. We  have added two sentences at the end of the second paragraph of the discussion.

Specific comments that would add to the manuscript are listed below.

  1. D) Editorial Comments: There many grammatical errors throughout the manuscript. A review by someone with English as a first language before submitting would have been helpful.

*It’s the 3rd time the manuscript has been reviewed by English speakers. In addition, we also revise our manuscript with a language software: Grammarly.

  1. E) Introduction: The introduction could be shortened with more concise writing. It is not necessary to list all of the predators previously reported, nor is it necessary to provide such detailed information on predator development time or fecundity.

*Our introduction is relatively short. It provides a good understanding of the system.

  1. F) Line 28: Is there a reference to document TPB is omnivorous? I have not heard this before.

*We are working on the omnivory of TPB as part of different projects that will be published soon.  There are also publications on the zoophagy of Lygus. We have added two to our text (Rosenheim & al. 2004; Solà & al. 2020).

  1. G) Line 50: The official common name of Orius insidiosus is the insidious flower bug, not minute pirate bug.

*The Anthocoridae are very frequently called minute pirate bug.

  1. H) Line 73: Please provide a reference that Orius feeds on pollen. Why (and how) would an insect with sucking mouthparts feed on pollen

*We have added a reference. Several Hemipterans feed on pollen.

  1. I) Insect Rearing: how long were these colonies maintained in the laboratory?

*More than a year. The population is constantly renewed with the capture of individuals in the field.

  1. J) Line 101-102: Not really sure what is being described here – what insect’s access was prevented and from what?

*Access to the water. We removed this part of the sentence to avoid confusion.

  1. K) Line 118: What other plants were regularly, and how were they cut.

*We have not kept a record of wild plants growing near strawberry mounds since it does not upgrade our experience.  We have kept the two most common abundant ones.

  1. L) Line 146: The design of the experiment needs to be explained to indicated that certain treatments were double-dipped – i.e., the same control plots (and probably 2 others) were used in the analysis of all 3 experiments. It’s confusing, because a total of 10 treatments across all 3 experiments are described, and with 12 replicates this equals 120 plots, but there were only 96 plots (line 114)

*We have added a sentence in the statistical analysis section to specify that the same control treatment was used in all models.

  1. M) Line 203: This pertains to the beginning of the each experiment results sections. The mean number of TPB per plant in the control is listed as 0.65. It’s not clear what this value represents – perhaps the average density across all sample dates?  If so, it is a meaningless value that has no relevance. What is more important is a description of TPB phenology across the study period.

*We have added details about this mean. We think this measure is interesting because it gives a point of comparison with the N. americoferus treatments. Figure 3 is also a good complement of information. We pooled all the TPB development stages because it gives a good estimation of the population without burdening the text. Moreover, all stages can cause damage and can be predated by N. americoferus.

  1. N) Line 309: what is anti-predatory behavior?

*We have added slight details about what anti-predatory behaviour is (e.g., escape, camouflage, use of refuge, etc…).

  1. O) Fig. 3-5: The color selection is not good for colorblind readers to distinguish treatments. In fact there’s really no need for different colors, as different shaped or filled and not filled data points along with dashed and solid lines could be used. Also are the vertical bars on data points standard error of mean?

*We have added shape to points, but we have kept the colors as before as it helps distinguishing the error bars. We have also specified that error bars are standard error in figure captions.

  1. P) The “Conclusion” is actually a Discussion and should be referred to as that. Also, there should be some discussion of how the results relate to application, and what next avenues of research are necessary. The last couple of sentences end rather abruptly with no conclusion.

*Ok, we have changed “conclusion” to “discussion”. The last paragraph focuses on the application of the method and an avenue for research to improve the approach.

Reviewer 2 Report

An interesting manuscript. My only major concern refers to the level of the pest where the study was conducted, which seems to be low, limiting the conclusions from the results. However, I understand how difficult it is to have a better setting in an experiment like this. Thus, I consider it would be important for the authors to address this issue in the manuscript.

I am also concerned with the possibility that the setup could have allowed the migration of the pest and the predators between plots. In the case of the predator, this could be even more critical, given the (apparently) low prey availability.

I have mentioned other suggestions in the file attached to this form.

Author Response

  1. A) All spelling, sentence changes and small details.

*Done (see comments in the PDF)

  1. B) I suggest including (in parentheses) the names of the corresponding order and families of these insects.

*We prefer not to include it since it makes the title longer and heavier.

  1. C) L17: How was the result affected by the low pest level in the field experiment? Please address this issue also in the discussion.

*The pest level in our experimental field was average compared to other years. In the control treatment, the level of TPB was largely above the economic threshold. Then, we don’t think it’s a specific point to address in the discussion.

  1. D) L27: It would be important to provide a little more detail on this aspect, as this is what justifies the conduction of this study. Where does it occur? What sort of damage it causes? What is the usual level of incidence? Etc.

*We don’t think those details are necessary for our text. Our experiment is not about why TPB is a major pest in strawberries, but about the biological control potential of two predators. Moreover, the damage caused by TPB is already well known.

L120: Any information about the approximate level of TPB and predators (including the species to be released) on the plants mentioned above (lines 116-117). Please refer to that here.

*This information is included in the results section.

L124 & L133 & L145: Please indicate how was this level selected

*The level was arbitrarily selected since, at the start of the experiment, we had no clue on the appropriate level required.

L149: Please provide details about this process, including the tapping itself, the location of the sampled plants within the plots, etc.

*As mentioned, the location of the plant was randomly selected. Any plant within the plot could be sampled. Some details were added about the method.

Why predators were not evaluated as well?? This would seem to complement so well the gathered data!

*The method used to monitor the prey was not as effective to monitor the predators. Hence, we only recorded a low number of predators. Therefore, we did not analyze these data.

L178 to L183: For this whole paragraph (and also for the following, which I suggest joining with this first paragraph), my suggestion is to compare treatments by (just) REFERRING TO THE NUMBER OF DEAD PREY in each treatment, instead of implying "comsumption" or "increase in prey mortality" for certain treatments. This is because actual predation was ASSUMED (most certainly correctly) NOT DETERMINED, and this can be reserved for "Discussion". Here in results, the actual evaluation referred to the number of dead prey (total of 15 minus live) at the end of the experiment. Thus, my suggestion is to refer just to the numbers of dead prey for each treatment (as indicated in Figure 1).

*We replaced the word “consumption” by “killed” or “number of dead prey”

L180: As mentioned above, my suggestion is for saying here that the number of dead N5 TPB was significantly higher in treatments with adults or N5 N. americoferus. Also for other comparisons in item 3.1, my suggestion is to use the same type of expression.

*Done.

L205 & L227 & L239: In which period?

*It is already mentioned at the very beginning of each paragraph.

As I mentioned before, I would suggest the use of a single decimal to express average populations, everywhere in the text.

*Done.

L206: It seems appropriate to make more explicity the statistical significance (or not) of the differences among low, medium and high release rates.

*We added the details (they are not different).

L207: Please specify clearly how this was calculated (which data: only control? All treatments? ...

*It’s a difference between the control and the N. americoferus treatment.

L229 : It should be "... did not significantly increase in either treatment". But I could not understand. If the control the mean number of TPB jumped from 0.3 in Aug 13-27 to 0.9 in Aug 27-Sept 17, then at least in this treatment the population increased, correct? Looking at Fig. 4, my impression is that increases were also observed for other treatments (though at a lower rate).

*It did not for the mentioned period (From August 27th to September 17th).

L248: For the field experiments, I suggest a thorough consideration concerning the possibility of the migration of both pest and predators between plots. This is extremely important, especially for the ability of the adults to fly.

*We already address this topic in the discussion. We underlined that we measured the local effect of the release predators.

Please refer to the actual levels of TPB in the field trials. Would the levels of occurrence be considered low, medium or high, in comparison to what is usual in grower fields? A low level of occurrence might not be interesting for a field study, limiting the conclusions that can be reached.

*The level was normal.

L286: Please re-word, as this seems to contradict what is mentioned in lines 266-267.

*We agree. The sentence was reworded accordingly.

L291: It would be interesting to discuss why those different levels did not produce statistically different results.

*We removed the sentence since these differences were not statistically significant.

L293: What leads to this suspicion?? How could this have affected the results obtained in the study? I think this deserve further consideration here, to define the limits of the conclusions of the study.

*We already underline that we “measured their local effect”, which involves the limits of the conclusions of the study.

L335: Article titles have almost all words starting in capital letters. This is not appropriate, at least for the scientific names they contain. Please correct.

*We used Zotero to automatically configure the reference. Hence, the style respects the journal’s instructions.

Reviewer 3 Report

The study entitled "The potential of Nabis americoferus and Orius insidiosus as bio logical control agents of Lygus lineolaris in strawberry fields" aims to determine the potential of N. americoferus as a biological control agent of L. lineolaris and evaluate the contribution of O. insidiosus, as a potential complementary agent to N. americoferus, in strawberry fields. The study is well structured, and the methods used were appropriate to achieve the proposed objectives. The study was complemented by laboratory and field studies. The manuscript is well-written and clear.

MINOR COMMENTS

Line 29:  add coma after "plant species"

Line 30: add a "and" before "buckwheat"

Line 53: add “the” before “presence”

Line 68: change “contributors” to “contributor”

Line 72: change “preys” to “prey”

Line 87: add “the” before field

Line 94: change “field” to “fields”

why the authors did not evaluate the variable sex on predation?

For better appealing esthetics, I recommend that the heading of each section must contain the full name of the insects. For example, change Line 123. “N. americoferus release rate”  for “Nabis americoferus release rate”

Lines: 177, 190, 201, 214 and 236, the italic in the species name is missing

Line 104 add a comma before having.

Line : the authors should add the photoperiod and temperature at which the laboratory test was performed; if the same as the “Insect rearing”, please indicate.

Line 105. change prey by another word. Such “insects” or even put the name of the insect.

Line 107 change "preys" to "prey"

Change: 177 and 214 "Nabis" for "Nabis americoferus" to maintain uniformity

Line 203. In "(± 0.10.)" remove the extra dot.

Author Response

Reviewer 3

MINOR COMMENTS

  1. A) Line 29:  add coma after "plant species"; Line 30: add a "and" before "buckwheat"; Line 53: add “the” before “presence”; Line 68: change “contributors” to “contributor”; Line 72: change “preys” to “prey”; Line 87: add “the” before field; Line 94: change “field” to “fields”; Line 104 add a comma before having; Line 105. change prey by another word. Such “insects” or even put the name of the insect; Line 107 change "preys" to "prey"; Lines: 177, 190, 201, 214 and 236, the italic in the species name is missing; Change: 177 and 214 "Nabis" for "Nabis americoferus" to maintain uniformity; Line 203. In "(± 0.10.)" remove the extra dot.

*All done.

  1. B) why the authors did not evaluate the variable sex on predation?

*It could in fact be an interesting addition to our research. We just did not measure it.

  1. C) For better appealing esthetics, I recommend that the heading of each section must contain the full name of the insects. For example, change Line 123. “N. americoferus release rate” for “Nabis americoferus release rate”

*Done.

  1. D) Line: the authors should add the photoperiod and temperature at which the laboratory test was performed; if the same as the “Insect rearing”, please indicate.

*Details added.